# Bi-Directional Axial Transmission measurements applied in a clinical environment

Jean-Gabriel Minonzio[1,2,3]*, Donatien Ramiandrisoa[4], Johannes Schneider[5], Eva Kohut[6], Melanie Streichhahn[6], Ulrik Stervbo[5,7], Rainer Wirth[8], Timm Henning Westhoff[7], Kay Raum[5], Nina Babel[5,7]

**1** Sorbonne Université, INSERM UMR S 1146, CNRS UMR 7371, Laboratoire d'Imagerie Biomédicale, Paris, France, **2** Escuela de Ingeniería Informática, Universidad de Valparaíso, Valparaíso, Chile, **3** Centro de Investigación y Desarrollo en Ingeniería en Salud, Universidad de Valparaíso, Valparaíso, Chile, **4** Bleu Solid, Pomponne, France, **5** Berlin-Brandenburg School for Regenerative Therapies, Charité - Universitätsmedizin Berlin, Germany, **6** Medical Clinic I, Marien Hospital Herne, Ruhr University, Bochum, Herne, Germany, **7** Center for Translational Medicine and Immune Diagnostics Laboratory, Medical Department I, Marien Hospital Herne, Ruhr University, Bochum, Herne, Germany, **8** Department for Geriatric Medicine, Marien Hospital Herne, Ruhr University Bochum, Herne, Germany

* jean-gabriel.minonzio@uv.cl

**Data Availability Statement:** Data are available at: https://osf.io/xy9qv/.

**Funding:** This work was supported by the EFRE. NRW program OsteoSys [EFRE-0800411 and

## Abstract

Accurate measurement of cortical bone parameters may improve fracture risk assessment and help clinicians on the best treatment strategy. Patients at risk of fracture are currently detected using the current X-Ray gold standard DXA (Dual XRay Absorptiometry). Different alternatives, such as 3D X-Rays, Magnetic Resonance Imaging or Quantitative Ultrasound (QUS) devices, have been proposed, the latter having advantages of being portable and sensitive to mechanical and geometrical properties. The objective of this cross-sectional study was to evaluate the performance of a Bi-Directional Axial Transmission (BDAT) device used by trained operators in a clinical environment with older subjects. The device, positioned at one-third distal radius, provides two velocities: VFAS (first arriving signal) and VA0 (first anti-symmetrical guided mode). Moreover, two parameters are obtained from an inverse approach: Ct.Th (cortical thickness) and Ct.Po (cortical porosity), along with their ratio Ct.Po/Ct.Th. The areal bone mineral density (aBMD) was obtained using DXA at the femur and spine. One hundred and six patients (81 women, 25 men) from Marien Hospital and St. Anna Hospital (Herne, Germany) were included in this study. Age ranged from 41 to 95 years, while body mass index (BMI) ranged from 16 to 47 kg.m$^{-2}$. Three groups were considered: 79 non-fractured patients (NF, 75±13years), 27 with non-traumatic fractures (F, 80±9years) including 14 patients with non-vertebral fractures (NVF, 84±7years). Weak to moderate significant Spearman correlations ($R$ ranging from 0.23 to 0.53, $p < 0.05$) were found between ultrasound parameters and age, BMI. Using multivariate Partial Least Square discrimination analyses with Leave-One-Out Cross-Validation (PLS-LOOCV), we found the combination of VFAS and the ratio Ct.Po/Ct.Th to be predictive for all non trau-matic fractures (F) with the odds ratio (OR) equals to 2.5 [1.6-3.4] and the area under the ROC curve (AUC) equal to 0.63 [0.62-0.65]. For the group NVF, combination of four param-eters VA0. Ct.Th, Ct.Po and Ct.Po/Ct.Po, along with age provides a discrimination model

EFRE-0800427, LS-1-1-019c]. Jean-Gabriel Minonzio is supported by Grant ANID / FONDECYT / REGULAR / 1201311 and ECOS 200061. The funders had no role in study design, data collection and analysis, decision to publish, or preparation of the manuscript. There was no additional external funding received for this study.

**Competing interests:** The authors state that they have no conflicts of interest.

with OR and AUC equals to 7.5 [6.0-9.1] and 0.75 [0.73-0.76]. When restricted to a smaller population (87 patients) common to both BDAT and DXA, BDAT ORs and AUCs are comparable or slightly higher to values obtained with DXA. The fracture risk assessment by BDAT method in older patients, in a clinical setting, suggests the benefit of the affordable and transportable device for the routine use.

## Introduction

Osteoporosis is associated with a pathological bone remodelling leading to an increased bone fragility and risk of fractures [1]. It remains a major public health problem worldwide, partly due to population aging, even if the fracture rate tends to decrease [2]. Therefore, it is still crucial to prevent the most severe fractures [3]. The current gold standard for detecting patients at risk of fractures is the Dual-energy X-ray Absorptiometry (DXA), assessing the areal or projected Bone Mineral Density (aBMD g.cm$^{-2}$), directly measured at the main fracture sites, i.e., hip and spine [4]. However, a majority of patients with fragility fractures have a T-score higher than the threshold defined by the World Health Organization (WHO), i.e., T-score = -2.5 [5]. Moreover, the accessibility to DXA is limited, usually restricted to larges cities, in numerous countries such as in Latin America [6]. That is why, several alternatives have been proposed.

Different X-rays technologies have emerged to assess bone quality, providing in particular cortical bone parameters. Indeed, because of its projection technique and low spatial resolution, DXA is not able to separate trabecular and cortical compartments. On the contrary, Quantitative Computed Tomography (QCT) has the ability to provide 3D bone volume, from which image processing techniques allow extracting material and geometrical properties such as cortical thickness (Ct.Th), cortical porosity (Ct.Po) and volumetric bone mineral density (vBMD g.cm$^{-3}$). While QCT has the advantage to be directly performed at the hip and spine [7], peripheral QCT (pQCT) is limited to peripheral sites such as tibia and radius [8]. Moreover, High resolution-pQCT (HR-pQCT) has a better resolution, allowing to directly image the larger pores, usually associated with resorption cavities [9, 10]. Similarly to QCT, Magnetic Resonance Imaging (MRI) has also been proposed as a DXA alternative, providing a MRI-derived porosity index in 3D volume [11].

Quantitative ultrasound (QUS) has been also proposed for clinical bone assessment, due to it advantages in term of cost, portability and sensitivity to both elasticity and geometry of the medium explored by the waves [12]. A first QUS measurement configuration corresponds to the transverse transmission, originally targeting cancellous bone at the heel [13], the earliest and best-validated clinical bone ultrasound device. Transverse transmission has also been applied at the one third distal [14] or distal [15] radius. A second configuration corresponds to back scattering or pulse-echo techniques, applied at the hip and spine [16], tibia [17] and radius [18, 19]. Moreover, note that echography has been recently enhanced to quantitative cortical bone imaging [20, 21]. Finally the last QUS configuration correspond to axial transmission (AT), with transducers aligned along the long bone axis [22]. The first measured clinical AT parameter was the velocity of the First Arriving Signal (VFAS also sometimes designated as speed of sound SOS), with frequencies of interest ranging from a few kHz [23], hundred of kHz [24] to a few MHz [25]. Its fracture discrimination ability has been shown similar or equivalent to DXA in numerous clinical studies since the 1990's [26]. Authors proposed to measure a second clinical AT parameter, the slowest velocity denoted fundamental flexural

guided waves (FFGW) [27] or the fundamental anti-symmetric Lamb mode A0 [28], with similar fracture discrimination abilities.

In order to provide performance beyond aBMD and ultrasonic velocities, complete AT signals could be studied. Indeed, the signal richness is linked to the waves guided by the cortical shell, originated in the reflections and interference along the propagation path [22]. Therefore, measured ultrasonic Guided Waves (GW) associated with an appropriate waveguide model have the potential to yield estimates of cortical parameters of clinical interest. For example, cortical thickness can be estimated using the FFGW [27] or combined material and geometrical cortical properties using all measured guided modes [29, 30]. Several signal processing approaches have been proposed to analyse GW measured with AT, such as the Radon transform for multichannel signals [31], dispersive Radon transform [32], adaptive array signal processing [33] or rotation invariant technique [34]. Likewise, Machine and Deep Learning approaches have been proposed to osteoporosis [35], guided waves [36] and both [37].

Using projection of waveguide models onto the singular vector basis of bidirectional axial transmission (BDAT) signals, concurrent identification of cortical thickness (Ct.Th) and porosity (Ct.Po) were validated on *ex vivo* human specimens [38, 39]. Then, in a first clinical study, assessment of Ct.Th and Ct.Po by BDAT were used to discriminate between (non traumatic) fractured and non-fractured patients [40]. In this pilot study, BDAT measurements were carried out by physicists using a simplistic guiding interface, based on the visual appreciation of the guided wave spectrum image [40]. The aim of this study is to test, in a second cross-sectional study, if the device could be easily used in a clinical environment, by hospital operators using an improved guiding interface relying on additional quantitative criteria. In addition to cortical thickness (Ct.Th) and porosity (Ct.Po), two velocities VFAS and VA0 are also measured in this study. Discrimation abilities of combination of these parameters with clinical factors will be tested using multivariate Partial Least Square discrimination analyses with Leave-One-Out Cross-Validation (PLS-LOOCV) and compared to DXA performance. Assessment of these BDAT parameters may improve the clinical identification of patients at high risk of fracture.

## Materials and methods

### Subjects

One hundred and nineteen patients (93 women, 26 men), aged from 41 to 95 years old, BMI from 16 to 47 kg.m$^{-2}$, were recruited from Marien Hospital and St. Anna Hospital (Herne, Germany) between August 2018 and February 2019. Exclusion criteria were: traumatic fracture ($n = 2$), ultrasonic measurement failure ($n = 11$), femoral DXA failure ($n = 10$) and vertebral DXA failure ($n = 9$). Due to the relatively small number of patients, two populations will be considered in the following: the complete successful BDAT measurement minus two traumatic fractures ($n = 106$), in order to explore BDAT fracture discrimination ability and the same group minus the 19 DXA failures ($n = 87$), in order to compare both techniques.

A written informed consent was provided by the subjects. The procedure of the study was in accordance with the Declaration of Helsinki. The OsteoSys study protocol was approved by the ethical committee of Ruhr University Bochum (no. 16–5714 approved on the 07.06.2016, including an amendment approved on the 24.01.2017) [41]. The information about non-traumatic fracture, including site and time, was collected in medical records. Three groups were created: a control group with patients without fracture (NF); patients with any non-traumatic fractures (F) divided in two subgroups: patients with vertebral fractures (VF) and patients with non vertebral fractures (NVF). Each patient of the fractured group presented only one non-traumatic fracture, i.e., there is no patient with multiple fractures.

## Measurements

Ultrasonic measurement were performed using the QUS device (Azalée, Paris, France) consisting in three custom-made parts. First, a 1-MHz bi-directional axial transmission (BDAT) probe adapted to forearm measurements (Vermon, Tours, France), is composed of a linear array of piezocomposite elements divided in three parts: one array of 24 receivers surrounded by two arrays of 5 transmitters each. The transmitter and receiver array pitches are respectfully equal to 1 and 0.8 mm. Second, an electronic device used to transmit, receive and digitize signals (Althaïs, Tours, France). The electronic device allows exciting each transmitter successively with a wideband pulse (170 V, 1-MHz central frequency). Third, a human machine interface (HMI), developed to display the spectrum of guided waves and provide the cortical parameters in quasi real-time (at a frame rate up to 4 Hz) and to guide the operator in finding during measurement the optimal position of the probe with respect to the main bone axis (BleuSolid, Paris, France).

Estimates of cortical thickness and porosity are obtained by searching the maximum position of the so-called *Proj* function. This approach is an extension of the signal processing applied to extract the experimental guided mode wave numbers from the maxima of the so-called *Norm* function [42], also denoted guided wave spectrum images [37]. In case of the *Norm* function, each pixel corresponds to the projection of attenuated plane plane into the basis of the reception singular vectors. Similarly, in case of the *Proj* function, each pixel corresponds to the projection of waveguide model, parametrized by cortical thickness and porosity. Typical examples of *Proj* functions (left column) along with their associated experimental guided modes corresponding to the maxima of the *Norm* functions (right column) are illustrated on Fig 1. Details about signal processing steps can be found in Refs. [38, 39]. In addition to Ct.Th and Ct.Po estimates, two ultrasonic velocities VFAS (first arriving signal) and VA0 (A0 Lamb mode) are also measured [28, 43]. Finally, five cortical parameters are obtained: two velocities VFAS and VA0; two inverse problem parameters Ct.Th and Ct.Po and additionally the ratio Ct.Po/Ct.Th.

A specific scanning methodology was carefully followed for measuring the patients. BDAT measurements were performed on a standardized region of interest (ROI), i.e., the center of the probe is placed at the lateral side of the one-third distal radius. Measurements were done on the non-dominant forearm. When the non-dominant forearm undergoes a recent fracture for example, measurements were done on the contralateral arm. The probe was placed in contact with the skin using ultrasonic gel for coupling (Aquasonic, Parker Labs Inc., Fairfield NJ, USA). The measurement protocol consists of four series of ten acquisitions. By means of the parameter values displayed in real-time by the HMI, once a correct position is found, a series of ten successive acquisitions are recorded without moving the probe. Each series corresponds to an intermediate repositioning of the probe. The final values of the identified waveguide parameters are set to the mean of the values of each successful series. The BDAT measurement on a patient typically lasts 10 to 15 minutes. Measurement were performed by clinical operators after a 2 day training session. In the previous pilot study, BDAT measurements were carried out by physicists using a first version of the guiding interface, based on the real time visualisation of the guided wave spectrum image, or *Norm* function [40]. In the updated version of interface, used in this study, several improvements have been implanted in order to make the measurement more robust, i.e., to ensure that the probe is correctly aligned with respect to the bone axis. First, real time measurement of the two velocities VA0 and VFAS are provided.Secondly, if these two values are obtained, the inverse problem is performed in real time, using a gradient approach calculation of the *Proj* function. On the contrary, if the two velocities are not obtained, the probe is considered as not correctly aligned and calculations are performed for the next probe position.

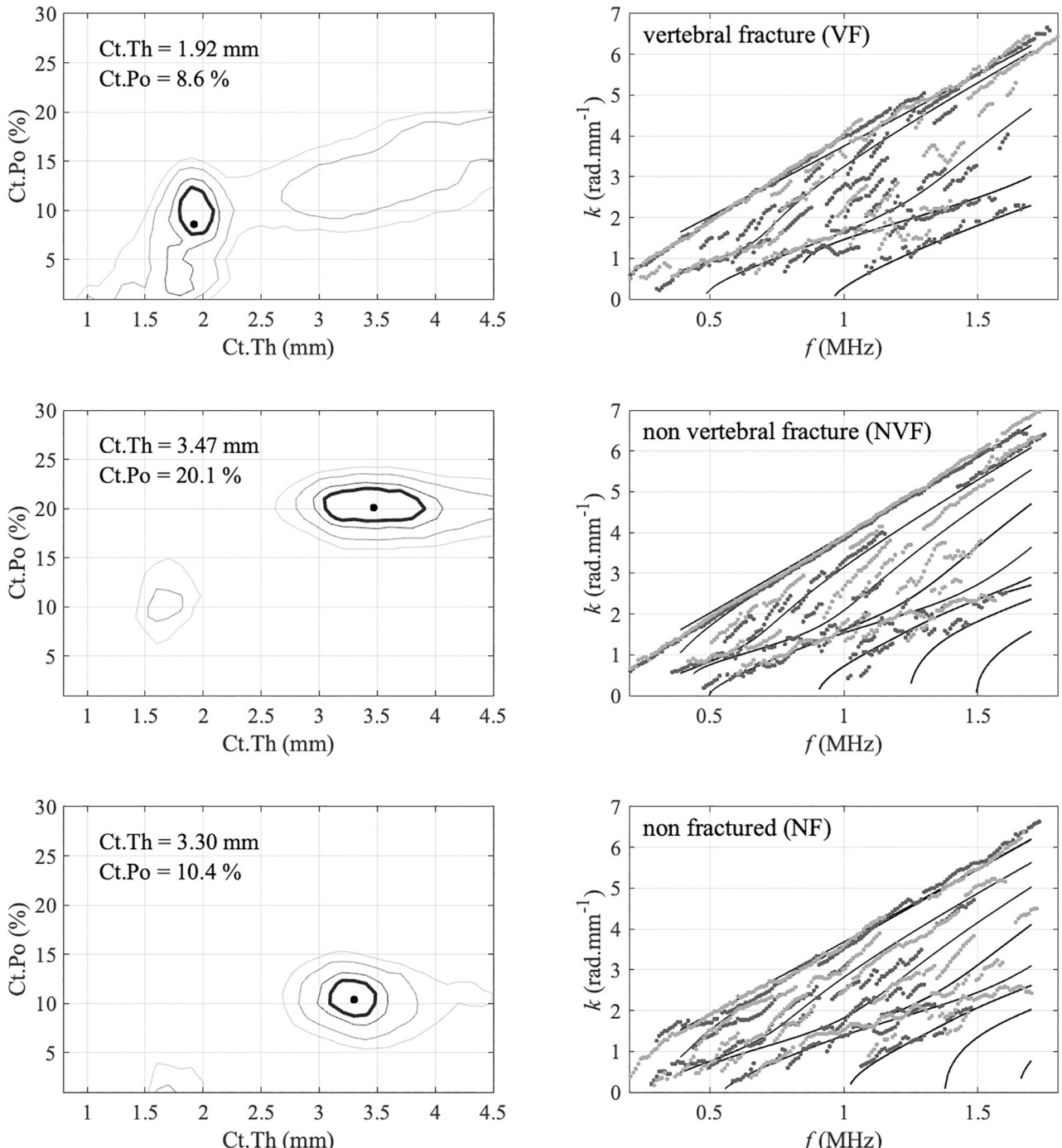

**Fig 1.** Typical examples of *Norm* (left) and *Proj* (right) functions measured *in vivo*: maximum in the (Ct.Th—Ct.Po) plane (right) corresponds to the best fitting wave guide model, shown with lines in the ($f$–$k$) plane (left), and provides estimates of cortical thickness and porosity.

Three DXA parameters are considered corresponding to measurements performed on the L1–L4 lumbar spine (aBMD spine), femoral neck (aBMD fn) and total femur (aBMD total). The two femoral aBMD values correspond to the mean of the values obtained for both femurs. If only one side was available, the retained aBMD corresponded to the existing value.

## Statistical analysis

Non parametric tests were used, and statistical results were considered significant for *p*-value values below 0.05. For each variable, a Wilcoxon–Mann–Whitney test was performed to determine whether the values were significantly different between the non-fractured group and any non-traumatic fractured group. Spearman's rank correlation analysis was used to compare estimates of ultrasonic parameters with age, BMI and aBMD values.

Following Armbrecht *et al.* [17], the fragility fracture discrimination performance of BDAT and DXA was assessed by means of multivariate Partial Least Square discrimination analyses with Leave-One-Out Cross-Validation (PLS-LOOCV) using the libPLS library [44]. Significant parameters were used to build a first set of discrimination models used to predict vertebral, non-vertebral, and all fragility fractures. A second set of models was also build using parameters associated with *p*-values lower than 0.01. In addition to DXA and/or BDAT parameters, other parameters were anthropometric data (weight, height, BMI), gender and age. The mean and standard error (SE) of the area under the curve (AUC) of the receiver operation characteristics (ROC), accuracy, sensitivity, and odds ratio (OR) with 95% confidence intervals (CI) were calculated. All statistical tests were performed using the Statistics Toolbox of Matlab R2021b (MathWorks, Natick, MA, USA).

Finally, in order to compare with classical statistical approach, odds ratios (ORs) were also calculated using binomial logistic regression analysis. ORs are expressed as increases in the estimated fracture risk per one standard deviation decrease. To estimate the sensitivity and specificity of the different parameters for the fracture discrimination, the receiver operator characteristic (ROC) curves were calculated. In addition, the area under the curve (AUC) was determined. Adjusted ORs and AUCs were computed with age, gender and BMI as covariates [45]. Combinations of BDAT, i.e., Ct.Th and Ct.Po as well as the ratio Ct.Po/Ct.Th, were also investigated. Results are provided in Supporting information.

# Results

## BDAT measurement failure

The ultrasonic measurement failed for 9.2% of the total cohort (11 out of 119). Fig 2 displays the failure rate as a function of BMI or age (up) together with the BMI and age distributions (down) for 106 patients. It can be observed that failure rate tends to increase with both age and BMI. However, increase with age appears to be more regular than increase with BMI: failure rate is constant and about 10% for BMI smaller than 32 kg.m$^{-2}$ while a a failure rate about 17% is observed for larger BMIs. Moreover, no association were observed between measurement failure and fractures.

## Patient characteristics

Among the 106 patients for whom the measurements were successful, 79 belongs to the control group without fracture (NF), 27 with non-traumatic fractures (F) corresponding to 13 patients with vertebral fractures (VF) and 14 with non vertebral fractures (NVF) associated with the following sites: 10 hip fractures, femoral neck (*n* = 7), pelvis (*n* = 2), pertrochanter (*n* = 1) as well as 4 others sites, humerus (*n* = 2), elbow (*n* = 1) and shoulder (*n* = 1). The patient characteristics are shown in Table 1.

As expected, age was significantly higher for fractured population (80 years) than for controls (75 years). The age difference is larger between non-vertebral fractures (85 years) and controls. Patients with vertebral fractures were found with similar age to controls (75 years). No significant differences were observed in BMI, height and weight between groups. Ct.Th

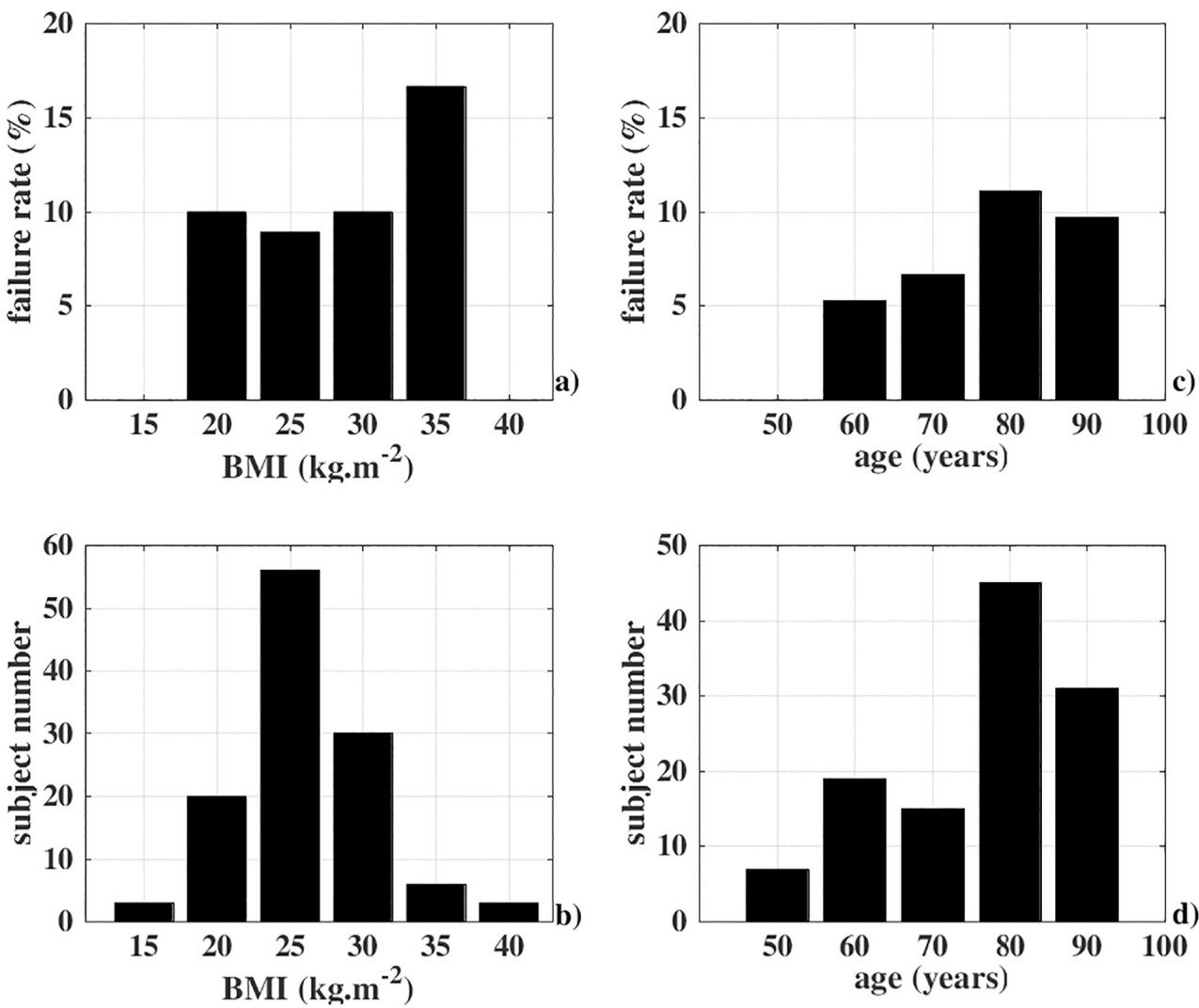

**Fig 2. BDAT measurement failure rate and distribution for BMI and age, for the first population (106 patients).**

**Table 1. Descriptive characteristics as mean and standard deviation for the first population (106 patients).**

|  | NF (N = 79) | F (N = 27) | VF (N = 13) | NVF (N = 14) |
|---|---|---|---|---|
| age (years) | 75.0 (13.1) | 79.7 (9.2) | 74.8 (9.4) | 84.2 (6.5)* |
| Height (*cm*) | 164.9 (8.6) | 164.7 (8.5) | 166.2 (9.0) | 163.7 (8.3) |
| Weight (*kg*) | 71.8 (18.4) | 69.2 (10.9) | 71.8 (11.3) | 66.7 (10.2) |
| BMI ($kg.m^{-2}$) | 26.2 (5.6) | 25.4 (3.4) | 26.1 (3.9) | 24.9 (3.0) |
| VFAS ($m.s^{-1}$) | 3993 (99) | 3949 (80)* | 3961 (87) | 3938 (75)* |
| VA0 ($m.s^{-1}$) | 1680 (70) | 1656 (70) | 1685 (68) | 1630 (63)* |
| Ct.Th (*mm*) | 2.7 (0.6) | 2.4 (0.7) | 2.5 (0.7) | 2.3 (0.7) |
| Ct.Po (%) | 10.0 (4.6) | 11.1 (4.3) | 9.3 (3.6) | 12.8 (4.2)* |
| Ct.Po/Ct.Th | 3.86 (1.74) | 4.84 (1.77)* | 3.92 (1.68) | 5.70 (1.42)*** |

BMI, body mass index; US ultrasound; Ct.Th, cortical thickness; Ct.Po cortical porosity; NF non fractured, F all non traumatic fractures VF vertebral fractures, NVF non vertebral fractures, $*p < 0.05$, $**p < 0.01$, $***p < 0.001$ *vs* non-fractured group.

**Table 2. Spearman's correlation coefficients *R* between BDAT parameters, age and BMI, for the first population (106 patients).**

|  | Ct.Th | Ct.Po | VFAS | VA0 | age | BMI |
|---|---|---|---|---|---|---|
| Ct.Th | - | 0.18 | 0.22 * | 0.41 *** | - 0.38 *** | 0.23 * |
| Ct.Po | 0.18 | - | - 0.21 ** | - 0.59 *** | 0.26 ** | -0.35 *** |
| VFAS | 0.22 * | - 0.21 ** | - | 0.31 ** | - 0.47 *** | 0.26 *** |
| VA0 | 0.41 *** | - 0.59 *** | 0.31 ** | - | - 0.53 *** | 0.42 *** |
| age | - 0.38 *** | 0.26 ** | - 0.47 *** | - 0.53 *** | - | -0.11 |
| BMI | 0.23 * | - 0.35 *** | 0.26 *** | 0.42 *** | -0.11 | - |

* $p < 0.05$,

** $p < 0.01$,

*** $p < 0.001$

was marginally lower in fractured groups (2.3—2.5 mm; $p = 0.10$) in comparison with the non-fractured group (2.7 mm). Ct.Po was higher in the F and NVF groups (11.1—12.8%) compared to the non-fractured group (10.0%), however the difference was only significant for the NVF group ($p = 0.03$). The ratio Ct.Po/Ct.Th was significantly higher in both F and NVF fractured groups (4.8—5.6) compared to the non-fractured group (3.9). The two ultrasonic velocities VFAS and VA0 are lower in the fractured groups. The VFAS difference is significant for both F and NVF groups while VA0 difference is significant only for NVF fractures. Spearman's correlation coefficients *R* between BDAT parameters, age and BMI are reported in Table 2. BDAT parameters are significantly correlated with age and BMI, with *R* ranging from 0.23 to 0.53. Ct.Th and Ct.Po as well as age and BMI are not correlated.

## Comparison between BDAT and DXA parameters

In order to compare BDAT performances with the current gold standard DXA, a second population of 87 patients was considered, taking into account failure cases of both techniques. First, Spearman's correlations were reported in Table 3. Almost all correlations were found significant, except aBMD spine with Ct.Th and VA0. Significant correlation coefficient *R* varies from 0.21 to 0.46. The largest coefficients, i.e., above 0.4, were observed for aBMD fn with Ct. Th/Ct.Po, VFAS and VA0. Estimation of the three aBMD from ultrasound Bi-Directional Axial Transmission and anthropometric parameters using PLS regression are shown in Fig 3, with $R^2$ ranging from 0.25 to 0.35.

**Table 3. Spearman's correlation coefficients *R* between BDAT and DXA parameters and the second population (87 patients).**

|  | aBMD total | aBMD fn | aBMD spine |
|---|---|---|---|
| Ct.Th | 0.21 * | 0.26 * | 0.09 |
| Ct.Po | - 0.28 ** | - 0.38 *** | - 0.21 * |
| Ct.Po/Ct.Th | - 0.39 ** | - 0.48 *** | - 0.24 * |
| VFAS | 0.46 *** | 0.45 *** | 0.39 *** |
| VA0 | 0.37 ** | 0.44 *** | 0.19 |

* $p < 0.05$,

** $p < 0.01$,

*** $p < 0.001$

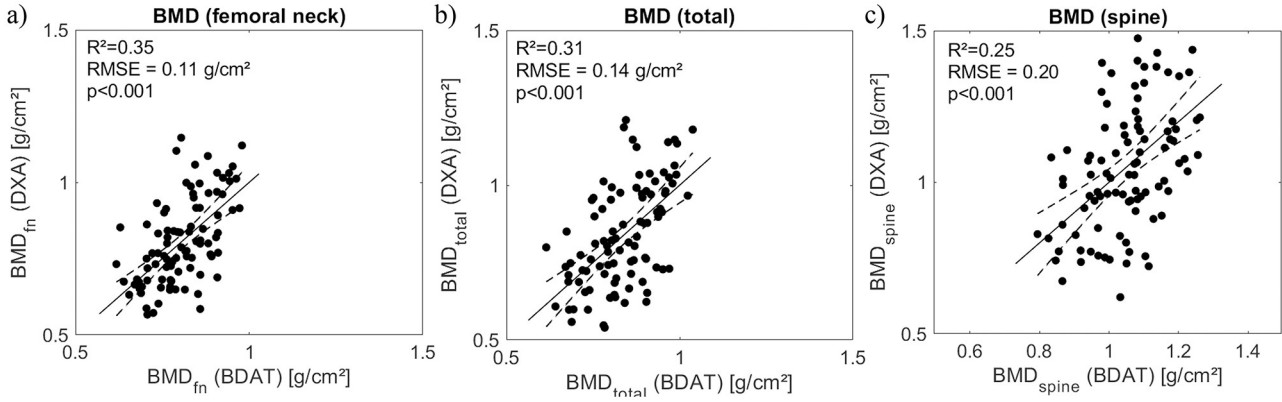

**Fig 3. Estimation of aBMD at femur neck (a), total femur (b) and spine (c) from ultrasound Bi-Directional Axial Transmission and anthropometric parameters using PLS regression.**

### Fracture discrimination

First, results of discrimination performance analyses obtained with PLS-LOOCV approach are considered and summarized in Table 4. The *p*-values of significant results ($p < 0.05$) are indicated in bold letters. Results for the vertebral fractures (VF group) are not shown as any combination parameters, either with BDAT or DXA, was found discriminant. For the first population (BDAT only, $N = 106$), all non traumatic fractures (F) and all significant parameters, combination of VFAS and Ct.Po/Ct.Th provided a moderate discrimination (AUC = 0.63, OR = 2.5). When restricted to the 14 non vertebral fractures (NVF), discrimination significantly improved to AUC equal to 0.75 and OR equal to 7.5. In this case, all five BDAT, along with age, are significant or very close (VFAS $p = 0.052$). Similar results are observed for the second population ($N = 87$), matched for both techniques. In case of all non traumatic fractures (F), moderate discrimination was observed for both techniques, with AUCs equals to 0.66 (BDAT and DXA) and ORs equals to 2.9 (BDAT) and 2.4 (DXA). As for the first population, better discrimination was observed for non vertebral fractures (NVF), with AUCs equals to 0.76 (BDAT) and 0.74 (DXA) and ORs equals to 16.4 (BDAT) and 7.1 (DXA). Both femoral aBMD were found significant contrary to spinal aBMD. When the two techniques were combined, an improvement was observed for the F group, with AUC and OR equals to 0.69 and 3.1, respectively. Considering the second set of models, build with parameters associated with $p < 0.01$, AUC and OR values tended to slightly increase. For example, for the NVF group of the second population, AUC increased from 0.76 to 0.78, while OR remained unchanged. For the NVF group, BDAT (AUC = 0.78) performed better than DXA (AUC = 0.74), while DXA (AUC = 0.67) slightly outperformed BDAT (AUC = 0.66) for F group. Combination of both techniques allowed a limited improved with AUC reaching 0.69 (F) and 0.79 (NVF). One can observe that the ratio Ct.Po/Ct.Th was the main parameter for the NVF group while the best discrimination performances were obtained using combination of parameters for the F group.

Secondly, results obtained with the classical method, i.e., logistic regression, are reported in S1 Table (first population) and S2 Table (second population) for the F and NVF groups. As before, the VF group is not reported due to the lack of significance. Systematic adjustment for cofounder variables age, BMI and gender was made for all the analyses due to the weak to moderate significant Spearman's correlations (*R* ranging from 0.23 to 0.53, $p < 0.05$ Table 2). Similar results compared to the first method (PLS-LOOC) were observed, e.g. moderate and

**Table 4. Discrimination obtained with PLS-LOOCV: $p$ values, odds ratios (OR) and areas under the ROC curve (AUC) for both BDAT and DXA techniques for the two populations (106 and 87 patients).**

| | N = 106 (79 NF) | | N = 87 (64 NF) | | | | | |
| --- | --- | --- | --- | --- | --- | --- | --- | --- |
| | BDAT | | BDAT | | DXA | | BDAT & DXA | |
| | 27 F | 14 NVF | 23 F | 13 NVF | 23 F | 13 NVF | 23 F | 13 NVF |
| Ct.Th | 0.053 | **0.035** * | 0.097 | 0.076 | - | - | 0.097 | 0.076 |
| Ct.Po | 0.246 | **0.032** * | 0.189 | **0.023** * | - | - | 0.189 | **0.023** * |
| VFAS | **0.040** * | 0.052 | **0.018** * | **0.024** * | - | - | **0.018** * | **0.024** * |
| VA0 | 0.135 | **0.015** * | 0.108 | **0.019** * | - | - | 0.108 | **0.019** * |
| Ct.Po/Ct.Th | **0.013** * | **<0.001** *** | **0.023** * | **<0.001** *** | - | - | **0.023** * | **<0.001** ** |
| age | 0.090 | **0.012** * | **0.025** * | **0.003** ** | **0.025** * | **0.003** ** | **0.025** * | **0.003** ** |
| height | 0.484 | 0.315 | 0.481 | 0.252 | 0.481 | 0.252 | 0.481 | 0.252 |
| weight | 0.978 | 0.645 | 0.622 | 0.373 | 0.622 | 0.373 | 0.622 | 0.373 |
| BMI | 0.482 | 0.370 | 0.659 | 0.381 | 0.659 | 0.381 | 0.659 | 0.381 |
| gender | 0.743 | 0.912 | 0.514 | 0.978 | 0.514 | 0.978 | 0.514 | 0.978 |
| aBMD fn | - | - | - | - | **0.003** ** | **0.002** ** | **0.003** ** | **0.002** ** |
| aBMD total | - | - | - | - | **0.007** ** | **0.009** ** | **0.007** ** | **0.009** ** |
| aBMD spine | - | - | - | - | 0.140 | 0.125 | 0.140 | 0.125 |
| models build with all significant parameters ($p < 0.05$) | | | | | | | | |
| **AUC** | **0.63** | 0.75 | **0.66** | 0.76 | 0.66 | **0.74** | **0.69** | 0.77 |
| CI | [0.62–0.65] | [0.73–0.76] | [0.64–0.68] | [0.73–0.78] | [0.65–0.68] | [0.72–0.76] | [0.66–0.71] | [0.75–0.80] |
| **OR** | **2.5** | 7.5 | **2.9** | 16.4 | 2.4 | **7.1** | **3.1** | 7.1 |
| CI | [1.6–3.4] | [6.0–9.1] | [1.9–4.0] | [14.3–18.5] | [1.4–3.4] | [5.5–8.7] | [2.1–4.1] | [5.5–8.7] |
| accuracy | 0.59 | 0.60 | 0.60 | 0.64 | 0.57 | 0.61 | 0.61 | 0.61 |
| sensitivity | 0.67 | 0.86 | 0.70 | 0.92 | 0.65 | 0.85 | 0.70 | 0.85 |
| specificity | 0.56 | 0.56 | 0.56 | 0.58 | 0.56 | 0.56 | 0.58 | 0.56 |
| models build with parameters $p < 0.01$ | | | | | | | | |
| **AUC** | - | **0.76** | - | **0.78** | **0.67** | 0.74 | 0.67 | **0.79** |
| CI | - | [0.76–0.76] | - | [0.77–0.79] | [0.65–0.69] | [0.72–0.76] | [0.65–0.69] | [0.77–0.81] |
| **OR** | - | **7.5** | - | **16.4** | **3.9** | 7.1 | 3.9 | **7.1** |
| CI | - | [6.0 9.1] | - | [14.3 18.5] | [2.8–4.9] | [5.5–8.7] | [2.8–4.9] | [5.5–8.7] |
| accuracy | - | 0.60 | - | 0.64 | 0.62 | 0.61 | 0.62 | 0.61 |
| sensitivity | - | 0.86 | - | 0.92 | 0.74 | 0.85 | 0.74 | 0.85 |
| specificity | - | 0.56 | - | 0.58 | 0.58 | 0.56 | 0.58 | 0.56 |

Reference category is non fractured (NF); CI confidence interval;

* $p < 0.05$;

** $p < 0.01$;

*** $p < 0.001$;

good discrimination for the F and NVF groups, respectively. Moreover, one can observed that the confidence intervals are systematically larger for the logistic regression compared to the PLS-LOOCV method. For example, in case of non vertebral fractures, the AUC for adjusted Ct.Po/Ct.Th is equal to 0.81 [0.65—0.90], whereas the PLS analysis provided a value of 0.76 [0.73—0.76] for the same patients of the first population (106 patients).

## Discussion

This study investigated the ability of the five BDAT parameters VFAS, VA0, Ct.Th, Ct.Po and the ratio Ct.Po/Ct.Th to discriminate retrospectively patients with non-traumatic fractures

from the control group, in a clinical environment. These parameters were estimated from measurements at the one-third distal radius using an ultrasonic guided wave technology, i.e., the Bi-Directional Axial Transmission (BDAT). In this second cross-sectional study, measurements were performed by clinical operators, in contrary to the pilot study, for which measurements were performed by laboratory staff. The discriminator values of BDAT parameters were assessed using multivariate Partial Least Square discrimination analyses with Leave-One-Out Cross-Validation (PLS-LOOCV) and a binomial logistic regression analysis. The main findings from this study were that VFAS and the ratio Ct.Th/Ct.Po were discriminant for all non-traumatic fractures (F). Discrimination was found better for non-vertebral fractures (NVF) with all BDAT parameters significantly or marginally discriminant. BDAT performance were found comparable (F) or slightly better (NVF) than the current gold standard DXA.

Our study demonstrate the clinical utility of bi-directional axial transmission measurements, corroborating previous BDAT study showing Ct.Po adjusted with age, BMI and cortisone, as discriminant parameter for all non-traumatic fractures as well as for subgroups hip, forearm and vertebral fractures [40]. However, in the pilot clinical study, the reference population was younger (NF 62 ± 7 years *vs* 73 ± 13 in this study) and no men were included. Moreover, the population was also larger with 201 included patients against 106 in the present study. The lack of sensitivity to vertebral fractures in this study may be partly explained by the fact that vertebral fractures are associated with trabecular bone [46] whereas the measurement site (one-third distal radius) is mainly cortical. In addition contrary to the pilot study, no significant age difference were observed between control and VF groups in the present study (Table 1).

An important observation was that the failure rate decreased from 20%, as observed in the pilot clinical study [40]), to 9% (11 out of 119), in this current study. This is likely due to the improvement of the guiding interface and acquisition protocol which were improved and adapted for trained operators. In the first version, the guiding feedback was visual and therefore more operator dependant. The operator was looking for the best guided wave spectrum image. The current version provides a real time numerical feedback about the probe alignment. In this study, the BDAT failure rate is lower than DXA one on the same population (17%, 20 out of 119). It can be noticed that the current BDAT failure rate is comparable to the usual DXA failure rate (10—11%) [47] and to failure rates of previous ultrasonic devices [43, 48]. It can be noticed that four patients corresponded to failure for both BDAT and vertebral DXA, however without manifest explanation.

A main advantage of the BDAT approach is to provide estimates of two physical parameters of cortical bone, thickness and porosity, which knowledge would potentially help the physician in his diagnosis [5]. Cortical thickness has been found discriminant of fracture in several previous studies [49]. For example, Long *et al.* showed in 2015 that the normalized cortical bone thickness (NCBT) estimated from the patient's hip DXA, discriminated hip fracture [50]. Likewise, Ye *et al.* demonstrated in 2020 that estimates of distal radius cortical thickness (DRCT), obtained from anteroposterior radiographs, can predict osteoporosis [51]. Likewise, cortical bone porosity is also known to be associated with fragility fractures [52]. Cortical bone parameters, derived from QCT or HR-pQCT, have been found discriminant for fragility fractures in numerous studies [7, 53]. For example, Ohlsson *et al.* shown that the cortical bone area predicts incident fractures independently of aBMD in older men [54]. Cortical bone porosity can also be evaluated from MRI ultrashort echo time (UTE) [11, 55]. In all previously cited methods, i.e., DXA, BDAT, QCT and MRI, the sub-millimetric spatial resolution does not allow to resolve the cortical pores. Therefore, cortical bone appears as a homogenized material, which elasticity can be parametrized with porosity [56]. Recently, it has been shown that cortical thinning and accumulation of large cortical pores in the tibia reflect local structural deterioration

of the femoral neck and can predict fragility fractures [10, 17]. The second HR-pQCT generation, with a lower resolution, is able to resolve the larger pores, usually associated with resorption cavities [17].

A first limitation of this study is the number of included patients, i.e., 106, and in particular the number of fractured patients, with numbers ranging from 13 to 27. It was observed that all five BDAT parameters were almost always significant or marginal. The statistical significance of each parameters as well as their combinations is expected to increase with larger patient groups. That is why, further studies would be necessary to strengthen the interest of the method. In particular, prospective studies would be of major interest. A second limitation is that fact that DXA measurements were performed on skeletal sites, i.e., spine and hip, that are different from the BDAT ones, i.e., one-third distal radius, which limits the comparisons between the two techniques. However, on can observe that the additive value in term of discrimination when combining BDAT and DXA (last column of Table 4). A third limitation concerns the comprehension of the failure cases. First, high values of soft tissue layer are known to limit BDAT measurement due to the difficulty to align the probe with the bone axis and as parallel as possible to the bone surface [43]. A second source of measurement failure is associated with the irregularity of the internal cortical bone layer due to trabecularisation [38, 57].

To overcome this last limitation, simulations of guided wave propagation using realistic cortical bone map should be helpful to gain insight about the impact of the waveguide irregularities on the guide mode propagation and their measurement [58]. Moreover, an improved guided interface could also help to diminish the failure rate. Tests are currently done with an interface showing the *Proj* function in a real time in addition to the *Norm* function (Fig 1). To this end, significant calculation speed up have been obtained [59]. In addition, improved detection of patients at risk of fractures may be also achieved by the incorporation of novel ultrasonic parameters. For example, cortical bone attenuation appears to be a factor of interest [60]. A recent study showed that this parameter could be experimentally estimated using Dictionary Learning and Orthogonal Matching Pursuit points of view [19]. Moreover, attenuation can be taken into account in the viscoelastic model and therefore into the waveguide model using a simplified homogenization model depending one additional temporal parameter $\tau^{matrix}$ [61]. Moreover, Machine learning approaches could also be used, in parallel or in addition to physical modeling, in order to improve the fracture discrimination [35, 37]. Deep Learning could be applied to guided wave spectrum or inverse problem images, following approaches already tested in medical ultrasound [62]. Finally, cortical bone parameters obtained with BDAT could be of interest beyond osteoporosis detection, for example for femoral stem stability monitoring [63] or other bone diseases [64].

## Conclusion

In summary, this study shows the potential of BDAT measurements to provide four *in vivo* parameters: cortical thickness and porosity estimates and two velocities VFAS and VA0, using a portable and non-ionizing device in a clinical environment.The fracture risk assessment by BDAT method in older patients, in a clinical setting, suggests the benefit of the affordable and transportable device for the routine use.

## Supporting information

**S1 Table. Discrimination obtained with logistic regression: Odds ratios (OR) and areas under the ROC curve (AUC) for BDAT technique on the first population (106 patients).** (PDF)

**S2 Table. Discrimination obtained with logistic regression: Odds ratios (OR) and areas under the ROC curve (AUC) for both techniques BDAT and DXA and the second population (87 patients).**
(PDF)

## Author Contributions

**Conceptualization:** Jean-Gabriel Minonzio, Donatien Ramiandrisoa, Johannes Schneider, Kay Raum.

**Data curation:** Jean-Gabriel Minonzio, Donatien Ramiandrisoa, Johannes Schneider, Eva Kohut, Melanie Streichhahn, Ulrik Stervbo, Kay Raum.

**Formal analysis:** Jean-Gabriel Minonzio, Donatien Ramiandrisoa, Johannes Schneider, Ulrik Stervbo, Kay Raum.

**Project administration:** Rainer Wirth, Timm Henning Westhoff, Nina Babel.

**Resources:** Rainer Wirth, Timm Henning Westhoff, Kay Raum, Nina Babel.

**Software:** Jean-Gabriel Minonzio, Donatien Ramiandrisoa.

**Supervision:** Jean-Gabriel Minonzio.

**Visualization:** Jean-Gabriel Minonzio.

**Writing – original draft:** Jean-Gabriel Minonzio.

**Writing – review & editing:** Jean-Gabriel Minonzio, Donatien Ramiandrisoa, Johannes Schneider, Eva Kohut, Melanie Streichhahn, Ulrik Stervbo, Rainer Wirth, Timm Henning Westhoff, Kay Raum, Nina Babel.

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
