## [Decision Letter · Decision Letter 0]

15 Aug 2022

PONE-D-22-03874Bi-Directional Axial Transmission measurements applied in a clinical environmentPLOS ONE

Dear Dr. Minonzio,

Thank you for submitting your manuscript to PLOS ONE. After careful consideration, we feel that it has merit but does not fully meet PLOS ONE’s publication criteria as it currently stands. Therefore, we invite you to submit a revised version of the manuscript that addresses the points raised during the review process.

Please address the concerns raised by the reviewers regarding rewriting and editing the manuscript and providing explanations on the previously published ultrasound methods that you cite in this manuscript, as well as providing a more detailed justification for the statistical methods and approaches. Please also consider the requirement for making your data available in a public data base.

We look forward to receiving your revised manuscript.

Kind regards,

John Leicester Williams, Ph.D.

Academic Editor

PLOS ONE

Journal Requirements:

 [This work was supported by the EFRE.NRW program OsteoSys [EFRE-0800411 and

EFRE-0800427, LS-1-1-019c]. Jean-Gabriel Minonzio is supported by Grant ANID /

FONDECYT / REGULAR / 1201311 and ECOS 200061. The funders had no role in study design, data collection and analysis, decision to publish, or preparation of the manuscript.”

“JGM is a cofounder of Azalée. The remaining authors state that they have no conflicts

of interest.”

We note that one or more of the authors are employed by a commercial company: Azalée

” This work was supported by the EFRE.NRW program OsteoSys [EFRE-0800411 and 305 EFRE-0800427, LS-1-1-019c]. Jean-Gabriel Minonzio is supported by Grant ANID / 306 FONDECYT / REGULAR / 1201311 and ECOS 200061. The authors would like to 307 thanks Azal´ee and BleuSolid for their support”

“This work was supported by the EFRE.NRW program OsteoSys [EFRE-0800411 and

EFRE-0800427, LS-1-1-019c]. Jean-Gabriel Minonzio is supported by Grant ANID /

FONDECYT / REGULAR / 1201311 and ECOS 200061. The funders had no role in study design, data collection and analysis, decision to publish, or preparation of the manuscript.”

5. We noticed you have some minor occurrence of overlapping text with the following previous publication(s), which needs to be addressed:

- https://ieeexplore.ieee.org/document/8579872

The text that needs to be addressed involves the Introduction.

In your revision ensure you cite all your sources (including your own works), and quote or rephrase any duplicated text outside the methods section. Further consideration is dependent on these concerns being addressed.

Reviewers' comments:

Reviewer's Responses to Questions

**Comments to the Author**

1. Is the manuscript technically sound, and do the data support the conclusions?

Reviewer #1: Partly

Reviewer #2: Yes

Reviewer #3: Yes

2. Has the statistical analysis been performed appropriately and rigorously? 

Reviewer #1: I Don't Know

Reviewer #2: I Don't Know

Reviewer #3: Yes

3. Have the authors made all data underlying the findings in their manuscript fully available?

Reviewer #1: No

Reviewer #2: No

Reviewer #3: Yes

4. Is the manuscript presented in an intelligible fashion and written in standard English?

Reviewer #1: No

Reviewer #2: Yes

Reviewer #3: Yes

5. Review Comments to the Author

Reviewer #1: The review comments to the authors can be found as an attachment.

Reviewer #2: The manuscript presents the results of a clinical study on the use of quantitative ultrasound for the detection of osteoporosis. The manuscript is well written and easy to understand. The manuscript focuses on the statistics extracted from the clinical study with limited details on the method the authors used. I understand that the method was already published in other papers but it would be useful to the reader to have a better description of the BDAT method. However, as a scientist with minimal knowledge in statistics, I can hardly judge the quality of the analysis of the results.

There are still a few points that I believe could improve the quality of the manuscript:

1. PLOS Data Policy requires author to make all data available. In this case, the authors say that the data is included in the manuscript. However, I believe that there would be value to the scientific community to be able to access the raw data (time traces) required to extract the bone parameters.

2. The current figure resolution is low. The quality should be improved before publication.

I finally noted a couple of typos:

line 46 - slower instead of slowest

line 299 - parameters

Reviewer #3: The authors evaluated Bi-Directional Axial Transmission (BDAT) device in clinical environment. Four in vivo parameters, including cortical thickness, cortical porosity, velocity of first arriving signal and velocity of A0 mode, are provided by the BDAT device used by the trained operators. The results suggest that the device is affordable and transportable for routine use. Generally, the paper is well organized. The results are solid and convincing. Therefore, the reviewer would recommend acceptance of this manuscript after minor revision. Below are some suggestions for the authors to improve the manuscript.

1. In page 3 line 91, what is the pitch size of the ultrasound probe?

2. In page 4 line 139, the author claim that the variable selection was found by subwindow permutation analysis using 1000 Monte Carlo samplings until a stable set was found. Can this part be explained more detailed?

3.In page 5 line 170, the author claims that the total is 9 hip fractures, but 8 femoral neck, 2 pelvis and 1 pertrochanter are 11 hip fractures. Please check the numbers again.

4. In page 2 line 43, some representative approaches in processing axial transmission signals can be considered to be listed before machine learning approaches, such as the Radon transform for multichannel guided waves signal processing and dispersive Radon transform method etc.

6. PLOS authors have the option to publish the peer review history of their article (what does this mean?). If published, this will include your full peer review and any attached files.

Reviewer #1: No

Reviewer #2: No

Reviewer #3: No

---

## [Author Response · Author response to Decision Letter 0]

23 Sep 2022

Dear Editor 

1) .tex files have been added 

2) and 3) The Competing Interests Statement is now

The authors state that they have no conflicts of interest.

4) the figure has been suppressed 

5) Table 6 is now cited in the text

Best Regards

Jean-Gabriel Minonzio

---

## [Decision Letter · Decision Letter 1]

4 Nov 2022

Bi-Directional Axial Transmission measurements applied in a clinical environment

PONE-D-22-03874R1

Dear Dr. Minonzio,

We’re pleased to inform you that your manuscript has been judged scientifically suitable for publication and will be formally accepted for publication once it meets all outstanding technical requirements.

Kind regards,

John Leicester Williams, Ph.D.

Academic Editor

PLOS ONE

Additional Editor Comments (optional):

Please note the PLOS Data policy requires authors to make all data underlying the findings described in their manuscript fully available without restriction, with rare exception.

Reviewers' comments:

Reviewer's Responses to Questions

**Comments to the Author**

1. If the authors have adequately addressed your comments raised in a previous round of review and you feel that this manuscript is now acceptable for publication, you may indicate that here to bypass the “Comments to the Author” section, enter your conflict of interest statement in the “Confidential to Editor” section, and submit your "Accept" recommendation.

Reviewer #2: All comments have been addressed

Reviewer #3: All comments have been addressed

2. Is the manuscript technically sound, and do the data support the conclusions?

Reviewer #2: Yes

Reviewer #3: Yes

3. Has the statistical analysis been performed appropriately and rigorously? 

Reviewer #2: I Don't Know

Reviewer #3: Yes

4. Have the authors made all data underlying the findings in their manuscript fully available?

Reviewer #2: No

Reviewer #3: Yes

5. Is the manuscript presented in an intelligible fashion and written in standard English?

Reviewer #2: Yes

Reviewer #3: Yes

6. Review Comments to the Author

Reviewer #2: The authors have addressed all my comments.

However, the data supporting the results will only be available on request.

Reviewer #3: The authors evaluated Bi-Directional Axial Transmission (BDAT) device in clinical environment. Four in vivo parameters, including cortical thickness, cortical porosity, velocity of first arriving signal and velocity of A0 mode, are provided by the BDAT device used by the trained operators. The results suggest that the device is affordable and transportable for clinical routine use. Generally, the paper is well organized. The results are solid and convincing. Therefore, the reviewer would recommend acceptance of this manuscript.

7. PLOS authors have the option to publish the peer review history of their article (what does this mean?). If published, this will include your full peer review and any attached files.

Reviewer #2: No

Reviewer #3: No

---

## [Editor Report · Acceptance letter]

22 Nov 2022

PONE-D-22-03874R1 

Bi-Directional Axial Transmission measurements applied in a clinical environment 

Dear Dr. Minonzio:

I'm pleased to inform you that your manuscript has been deemed suitable for publication in PLOS ONE. Congratulations! Your manuscript is now with our production department. 

Kind regards, 

on behalf of

Dr. John Leicester Williams 

Academic Editor

PLOS ONE